# GAIR : GUI Automation via Information-Joint Reasoning and Group Reflection

## Abstract

Building AI systems for GUI automation task has attracted remarkable research efforts, where MLLMs are leveraged for processing user requirements and give operations. However, GUI automation includes a wide range of tasks, from document processing to online shopping, from CAD to video editing. Diversity between particular tasks requires MLLMs for GUI automation to have heterogeneous capabilities and master multidimensional expertise, raising problems on constructing such a model. To address such challenge, we propose GAIR : **G**UI **A**utomation via **I**nformation-Joint Reasoning and Group **R**eflection, a novel MLLM-based GUI automation agent framework designed for integrating knowledge and combining capabilities from heterogeneous models to build GUI automation agent systems with higher performance. Since different GUI-specific MLLMs are trained on different dataset and thus have different strengths, GAIR introduced a general-purpose MLLM for jointly processing the information from multiple GUI-specific models, further enhancing performance of the agent framework. The general-purpose MLLM also serves as decision maker, trying to execute a reasonable operation based on previously gathered information. When the general-purpose model thinks that there isn't sufficient information for a reasonable decision, GAIR would transit into group reflection status, where the general-purpose model would provide GUI-specific models with different instructions and hints based on their strengths and weaknesses, driving them to gather information with more significance and accuracy that can support deeper reasoning and decision. We evaluated the effectiveness and reliability of GAIR through extensive experiments on GUI benchmarks.

## 1 Introduction

GUI automation task, aiming at utilizing AI systems for automating GUI operations, would bring further convenience for people's daily work, thus becoming an attractive research field. With the rapid development of Large Language Models (LLMs) and Multimodal Large Language Models (MLLMs), the capabilities of AI models have reached the level of human intelligence, making it possible to construct AI systems for GUI automation. Researchers have found that using MLLMs for GUI automation task can economize computational cost (Xu et al., 2024), enhance generalization (Liu et al., 2025b) and reduce model hallucination (Meng et al., 2024) and thus reaching better performance (Kil et al., 2024). Therefore, the outstanding capabilities for processing complex semantic information and contextual information and the architecture design that can simultaneously process both textual and visual input have made MLLMs an appropriate choice for constructing AI systems for GUI automation. However, GUI automation task is continually expanding, involving more and more real-world scenarios, where applications, GUI pages and action spaces in each scenario can be completely different, thus requiring different MLLMs to master such capabilities and achieve GUI automation in each scenario and scope of task. For instance, general GUI automation models like AGUVIS (Xu et al., 2024) and Ferret-UI (Li et al., 2025) does well on web and mobile applications that are common in people's daily life, but inefficient on various tasks in vertical fields; Assistant systems designed for specific scenario or discipline such as CAD-Assistant (Mallis et al., 2024) would be able to handle tasks within the designed scope but unable to process various tasks in people's daily life. In addition, as demonstrated in Fig. 1, the process of GUI automation tasks involves multiple sub-tasks such as GUI page information extraction, information-processing-based decision

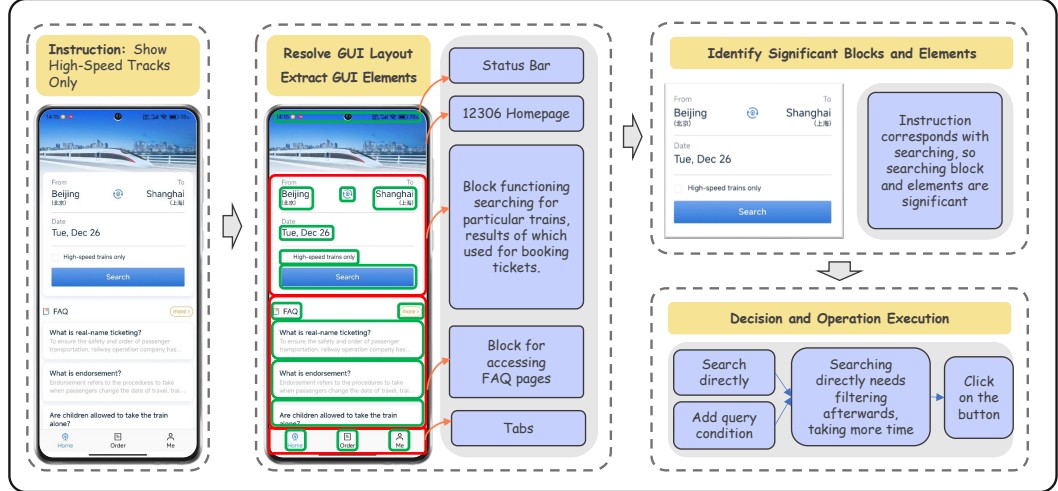

Figure 1: Demonstration of sub-tasks involved in the process of GUI automation task

and precise operation generation, bringing higher complexity. Therefore, to build an efficient and reliable agent system that is capable for most GUI automation tasks, multiple MLLMs need to be leveraged. In this paper, we would explore the way to combine the capabilities of multiple GUI-specific MLLMs to construct a GUI automation agent system. By making full use of the models' capabilities, the agent system would be able to cover tasks in more scenarios and disciplines.

The diversity of GUI automation tasks and the complexity of sub-tasks involved in the completion process of GUI automation tasks introduce multifaceted challenges on integrating capabilities of multiple models to construct a high performance GUI automation agent system, the significant ones of which we identified are as follows. i) **Information Alignment and Discrimination.** Different GUI-specific models have different strengths and weaknesses, thus showing different competences on different tasks. To gather valid and reliable information through these models, the agent system needs to efficiently integrate such information and to tell which model is more reliable as the information source of current task. ii) **Conflicting Decision and Operation Precision.** Generating precise GUI operation is a task that GUI-specific models would do better on, while efficiently integrating information and making reasonable decisions accordingly is not a strength of GUI-specific models that have been finetuned on GUI data. To construct efficient and reliable GUI automation agent system, mechanisms should be designed to address such problem. iii) **Error Avoidance and Correction.** Incorrect or inaccurate operation execution would result in more time latency, and sometimes directly cause the failure of the task. Therefore, enabling the agent system to detect incorrect or inaccurate operation before the operation gets executed would further enhance the performance of the agent system.

In this paper, we introduce GAIR : **G**UI **A**utomation via **I**nformation-Joint Reasoning and Group **R**eflection. GAIR is an agent framework that integrates multifaceted capabilities from multiple models to accomplish GUI automation task with a higher performance. GAIR employs multiple GUI-specific models to extract information from GUI pages, leveraging GUI-specific capabilities from multiple training dataset that cover wider range of tasks. To efficiently integrate information and tell the reliability of each model, GAIR leverages general-purpose MLLM with outstanding capabilities on processing complex contextual and semantic information. In addition, we employ multi-turn dialogue to enable step-by-step reasoning of the general-purpose MLLM. Such mechanism could increase the effectiveness of information integration and discrimination of the agent system, additionally balancing the reliability of decision and the precision of operation execution simultaneously. For error avoidance, we introduce group reflection mechanism to further enhance GAIR framework. GAIR can decide to transit into reflection state when information is not sufficient, when the general-purpose model would give instructions to GUI-specific models to drive them to gather more useful information, and make reasoning and decision based on new information again.

We validated GAIR's effectiveness and reliability with evaluations on benchmarks involving GUI automation tasks on web, desktop and mobile platforms. On UI-I2E-Bench (Liu et al., 2025a) and ScreenSpot (Cheng et al., 2024), GAIR achieves success rate of 78.4% and 91.0% respectively, surpassing the current state-of-the-art 72B model (with success rate of 76.3% and 89.5% respectively) while employing 7B MLLMs only. Such results demonstrate that the group reflection with information-joint reasoning mechanism successfully enhances GAIR's capabilities on gathering and fully processing information about GUI pages and tasks, making reasonable decisions and executing precise operations.

## 2 RELATED WORKS

**GUI Automation Task.** GUI automation task is a task that requires AI systems to process GUI environment, understand user requirement and give appropriate GUI operation. Various datasets has been constructed for training and evaluating the abilities of AI systems on accomplishing GUI automation (Deng et al., 2023; Cheng et al., 2024; Gao et al., 2024; Lu et al., 2024; Zhang et al., 2024a; Kapoor et al., 2024; Rawles et al., 2025; Xu et al., 2025). Completing such task requires AI systems to have multidimensional capabilities, such as perceiving the information, deciding what to do, and executing the operation precisely. Multiple tasks are proposed to better train and evaluate the capabilities required for accomplishing GUI automation, such as GUI referring tasks that requires AI systems to process GUI pages and output the description of certain element or information based on the prompt (Lù et al., 2024; Pan et al., 2024; Wang et al., 2024; Tian et al., 2025), and GUI grounding tasks that requires AI systems to give the location of the required GUI element based on a description (Li et al., 2020; 2021; Wang et al., 2021; Venkatesh et al., 2022). Training on such tasks can help enhancing models' capabilities on processing GUI pages and understanding user requirements, and constructing a model for GUI automation requires such capabilities. By synergizing multiple models, we further enhance such processing and understanding capabilities of AI agent systems for GUI automation tasks.

**MLLM for GUI Automation Task.** As the development of LLMs and MLLMs has shown great advancements on improving the ability to perceive and understand complex semantic and contextual information, leveraging LLMs or MLLMs for GUI automation tasks attracted research efforts. LLMs cannot process screenshot images of GUI pages directly, which makes it more efficient and convenient for MLLMs to process and complete GUI automation task. Some works thus use MLLMs with auxiliary tools to build GUI automation agents (Sun et al., 2022; Yan et al., 2023; Zheng et al., 2024; Ding, 2024; Zhou et al., 2024; Bonatti et al., 2024; Meng et al., 2024; Lu et al., 2024). However, due to the distribution misalignment between GUI screenshots and images used for training general-purpose MLLMs, the perception capabilities of these agent systems highly rely on the auxiliary tools, which has far less knowledge compared to MLLM. Therefore, most works chose existing MLLM backbone such as Qwen (Bai et al., 2025) and LLaVA (Liu et al., 2023) and fine-tune the models to construct an MLLM for GUI automation (Zhang et al., 2023; Wu et al., 2024; Baechler et al., 2024; Chen et al., 2024; Lu et al., 2024; Liu et al., 2025b; Gou et al., 2025; Xu et al., 2025). Existing and synthesized data are used to train and optimize these models. In addition, some of the works introduce extra modules into the model to further enhance the GUI referring and grounding abilities. Some of them employ high-resolution vision encoders or additional special encoders for GUI processing (Nong et al., 2024; Hong et al., 2024; Zhang et al., 2024b), some others introduce image clipping mechanisms to enable perception of any-resolution screenshot images (You et al., 2024; Ge et al., 2024; Lin et al., 2025; Yang et al., 2025; Li et al., 2025). However, finetuning and optimizing the model to enhance GUI abilities may follow forgetting on other general abilities, where our combination of heterogeneous models can better reserve all capabilities required for GUI automation tasks.

## 3 METHOD

In this section, we present the GAIR framework in detail. As illustrated in Figure 2, GAIR decomposes the GUI automation process into a few core components. To fulfill these components effectively, GAIR leverages an ensemble of MLLMs with heterogeneous capabilities, strategically harnessing the unique strengths of each model. In the following subsections, we elaborate on how these models' capabilities are used to fulfill the components and accomplish each stage of the task.

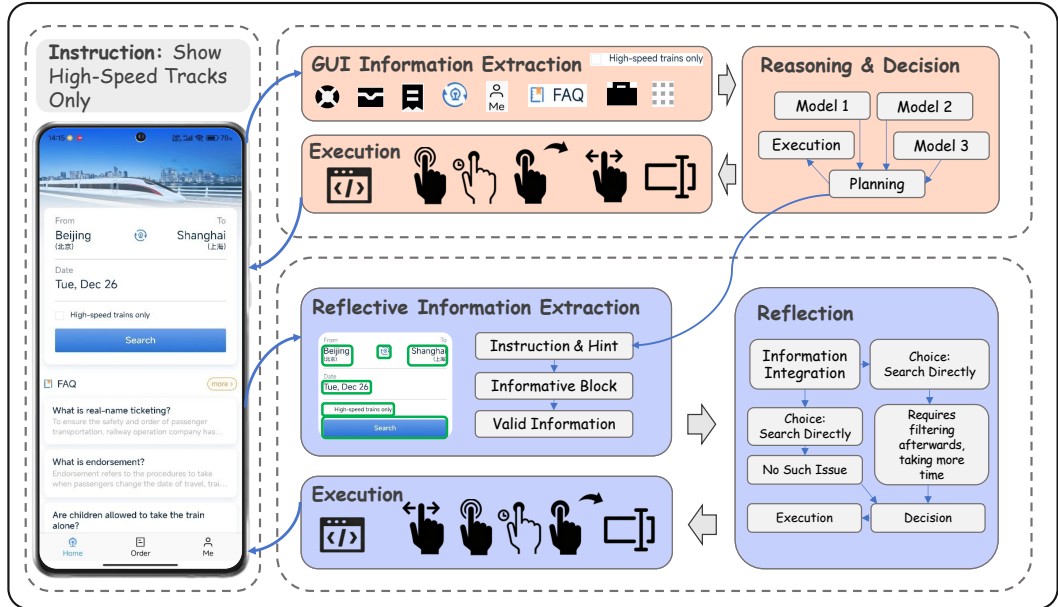

Figure 2: System overview of GAIR

## 3.1 OBSERVATION

For GUI automation agents, observation — that is, the accurate extraction of semantically meaningful information from graphical user interfaces — constitutes a foundational capability. Inaccurate or incomplete observations invariably lead to cascading failures in downstream subtasks.

The rapid advancement of MLLMs has enabled AI systems to achieve human-level intelligence and to process visual and textual inputs simultaneously. However, applying general-purpose MLLMs to extract structured information from GUI screenshots remains a significant challenge due to the stark divergence between natural images and GUI layouts. Natural images typically feature a limited number of salient objects occupying substantial portions of the image area. In contrast, GUI screenshots are densely populated with numerous interactive elements, any of which may assume critical importance depending on the specific task. Moreover, task-relevant GUI elements that are of remarkable significance may occupy less than 1% of the screen area, making it challenging for general-purpose MLLMs to detect or ground with sufficient precision. To ensure robust and reliable perception in GUI automation agent systems, it is imperative to employ GUI-specific MLLMs that are fine-tuned or optimized for GUI understanding. However, the GUI-specific capabilities of GUI models are derived from their training data, meaning these capabilities are inherently constrained by the scope and distribution of that data. In other words, the GUI-specific capabilities of the models are determined by the training data to some extent. Specifically, the GUI-related competencies of GUI-specialized MLLMs reflect the distribution of their training datasets, rather than encompassing the full diversity of real-world GUI screenshots. To ensure the agent system can reliably extract sufficient and accurate information from GUI environments across a broader range of tasks, employing multiple GUI-specialized MLLMs for observation is sufficient.

Therefore, GAIR integrate an ensemble of GUI-specific MLLMs to jointly process each GUI screenshot in conjunction with the user's task specification. These models collaboratively extract, interpret, and articulate necessary information, thereby providing a comprehensive and reliable observation for subsequent process.

## 3.2 REASONING AND DECISION

After acquiring sufficient information from GUI environment, the agent needs to integrate the information, make sufficient reasoning and decisions to select appropriate operation from current action

space. The capabilities of GUI automation agents on reasoning and operation selecting directly determines the efficiency and reliability of the agent system.

While GUI-specific models feature outstanding abilities on GUI referring, grounding and navigation, general-purpose MLLMs has more comprehensive world knowledge and stronger natural language processing capabilities. Since the GUI-specific models have extracted significant information from the observation of GUI page and expressed the information in textual form, general-purpose MLLMs would be able to directly process the textual GUI-specific information efficiently and reliably. General-purpose MLLMs would do better on processing complex semantic and contextual information in natural language compared to GUI-specific MLLMs. In addition, richer world knowledge of general-purpose MLLMs would help with reasoning and decision making in a lot of ways. For example, in certain cases, with such knowledge, general-purpose MLLM can directly identify the correct operation based on the GUI page and user requirement, while only the grounding ability of GUI-specific MLLMs are required for generating the precise operation directly; in particular tasks, general-purpose MLLM can rely on its reasoning and generalization capabilities to analyze the layout and functions of out-of-distribution GUI pages for GUI-specific MLLMs; furthermore, general-purpose MLLMs can try to tell if the information from one of the specific models are correct by leveraging its own knowledge or comparing it with information from the others, further enhancing the performance of the agent system. Some works have shown that strong MLLMs are capable of planning and decision making in GUI automation task (Yan et al., 2023; Zheng et al., 2024). Therefore, it would be feasible to utilize a general-purpose MLLM to integrate the information gathered by GUI-specific MLLMs and make operation selection.

In GAIR, we leverage a general-purpose MLLM for integrating information, reasoning, decision making and operation selection. The strong capabilities on synthesizing and analyzing complex semantic and contextual information of general-purpose MLLMs enables such utilization and would further enhance the efficiency and reliability of the whole agent system.

### 3.3 GROUP REFLECTION

In some cases, one turn of information extracting and decision making might be unable to complete the task or current step due to the complexity of the task or unhelpful collaboration. To address such challenge and construct a better GUI automation agent system, we introduce group reflection mechanism in GAIR framework, improving the capabilities and robustness of the agent system.

When the general-purpose MLLM for reasoning and decision thinks that the gathered information is not sufficient for making proper decision and select correct operation, the agent system will decide to transit into reflection state, and a series of internal actions would be taken. The general-purpose MLLM would give different instructions and hints to different GUI-specific models based on the information they gathered, indicating what kind of information is required and considering their strengths and weaknesses accordingly. As the GUI-specific models receive further instructions and hints, they would try to process the GUI page following the instructions and hints, trying to extract and analyze more useful information from current GUI status. The general-purpose MLLM would accept such information from the GUI-specific models that has been converted into textual form immediately afterwards, and reuse its capabilities on synthesizing and analyzing complex contextual and semantic information to fully process the responses from GUI specific models, trying deeper reasoning and resolving to ensure a more reliable decision. Such mechanisms enables the agent system to make full use of both the GUI-specific capabilities and the complex language processing abilities for a more reliable reasoning and a more precise execution, thus further enhancing the performance of the agent system.

## 4 EXPERIMENT

### 4.1 IMPLEMENTATION DETAILS

#### 4.1.1 DATASETS

We evaluated our method on two benchmarks, ScreenSpot (Cheng et al., 2024) and UI-I2E-Bench (Liu et al., 2025a). ScreenSpot is a benchmark containing more than 1,200 samples, designed for assessing GUI automation agent systems' capabilities on giving the precise location of the desired

Table 1: Quantitative results on UI-I2E-Bench benchmark. Results marked in **bold** represents the best performance, while results underlined represents the second best performance. Scores of baselines are taken from papers leaderboards.

| Method | Platform | | | Element Type | | | | | Overall |
|---|---|---|---|---|---|---|---|---|---|
| | Web | Desktop | Mobile | Button | Icon | Dropdown | Input | Toggle | |
| OmniParser | 30.8 | 45.5 | 67.6 | 68.4 | 60.5 | 65.9 | 58.9 | 26.9 | 53.1 |
| AGUVIS-7B | 45.1 | 47.6 | 60.3 | 60.2 | 56.3 | 74.2 | 54.7 | 35.7 | 53.2 |
| OS-Atlas-7B | 52.2 | 48.9 | 68.1 | 69.1 | 58.7 | 80.3 | 70.1 | 32.3 | 58.6 |
| UI-TARS-7B | 56.5 | 58.0 | 65.7 | 66.5 | 63.3 | 75.3 | 60.4 | 51.4 | 61.4 |
| InfiGUI-R1-3B | 71.7 | 57.2 | 78.2 | 71.6 | 67.5 | 82.6 | 74.2 | 60.4 | 69.7 |
| UGround-V1-7B | 70.8 | 65.7 | 73.5 | 72.9 | 62.9 | 83.7 | 75.4 | 63.5 | 70.3 |
| UI-TARS-1.5-7B | 79.5 | 68.8 | 74.1 | 76.6 | 71.7 | 82.0 | 75.3 | 66.3 | 73.2 |
| UI-TARS-72B | 77.1 | 69.8 | 75.5 | 78.8 | 75.2 | 80.9 | 73.9 | 66.0 | 73.7 |
| UGround-V1-72B | 74.7 | **74.6** | 78.2 | 79.6 | **75.5** | **93.3** | 74.5 | 68.7 | 76.3 |
| **Ours** | **82.2** | 72.8 | **81.1** | **81.8** | 74.1 | 89.9 | **78.6** | **74.5** | **78.4** |

Table 2: Quantitative results on ScreenSpot benchmark. Results marked in **bold** represents the best performance, while results underlined represents the second best performance. Scores of baselines are taken from papers leaderboards.

| Method | Mobile | | Desktop | | Web | | Overall |
|---|---|---|---|---|---|---|---|
| | Text | Icon | Text | Icon | Text | Icon | |
| AGUVIS-7B | 95.6 | 77.7 | 93.8 | 67.1 | 88.3 | 75.2 | 83.0 |
| OS-Atlas-7B | 93.8 | 79.9 | 90.2 | 66.4 | **92.6** | 79.1 | 85.1 |
| UGround-V1-7B | 93.0 | 79.9 | 93.8 | 76.4 | 90.9 | 84.0 | 86.3 |
| InfiGUI-R1-3B | **97.1** | 81.2 | 94.3 | 77.1 | 91.7 | 77.6 | 87.5 |
| AGUVIS-72B | 94.5 | 85.2 | 95.4 | 77.9 | 91.3 | 85.9 | 88.4 |
| UGround-V1-72B | 94.1 | 83.4 | 94.9 | **85.7** | 90.4 | **87.9** | 89.4 |
| UI-TARS-7B | 94.5 | 85.2 | **95.9** | **85.7** | 90.0 | 83.5 | 89.5 |
| **Ours** | 96.0 | **88.2** | **95.9** | 84.3 | 91.8 | 86.4 | **91.0** |

interaction position based on particular GUI screen status and user requirement. ScreenSpot benchmark includes samples on web, mobile and desktop platforms. UI-I2E-Bench is also designed for assessing the capabilities of GUI automation agent systems on giving the precise position of the proper interaction based on GUI screenshot and user intent. UI-I2E-Bench contains 1,477 samples with a more detailed annotation of platform type, element type, etc., enabling more detailed diagnosis on performance of GUI automation agents. These benchmarks offer comprehensive evaluation of the agent system's capabilities on different platforms, different types of elements and different types of instructions.

### 4.1.2 IMPLEMENTATION DETAILS

As demonstrated above, the GAIR framework contains multiple GUI-specific MLLMs for information extraction and a general-purpose MLLM for reasoning and decision. Qwen2.5-VL-7B Bai et al. (2025) is the general-purpose MLLM used in the agent system, serving as information integrator and decision maker. In our experiment, we leverage the following GUI-specific MLLMs: UI-TARS-7B-DPO (Qin et al., 2025), InfiGUI-R1-3B (Liu et al., 2025c) and UGround-V1-7B (Gou et al., 2025). UI-TARS-7B-DPO uses Qwen2-VL-7B as backbone, and is trained on extensive GUI data consisting of approximately 50B tokens, capable of GUI automation task and various subtasks. InfiGUI-R1-3B uses Qwen2.5-VL-3B as backbone, and is trained on approximately 32K high-quality GUI

Table 3: Ablation study of GAIR components, where SM indicates using optimal single model only, MM indicates using multiple model with joint information reasoning but without group reflection, and GAIR indicates using multiple model with joint information reasoning and group reflection, which is the whole GAIR pipeline.

| Method | Platform | | | Element Type | | | | | Overall |
|--------|-----|---------|--------|--------|------|----------|-------|--------|---------|
| | Web | Desktop | Mobile | Button | Icon | Dropdown | Input | Toggle | |
| SM | 70.8 | 65.7 | 73.5 | 72.9 | 62.9 | 83.7 | 75.4 | 63.5 | 70.3 |
| MM | 80.6 | 70.9 | 79.1 | 80.3 | 74.5 | 89.9 | 75.7 | 69.4 | 76.5 |
| GAIR | 82.2 | 72.8 | 81.1 | 81.8 | 74.1 | 89.9 | 78.6 | 74.5 | 78.4 |

Table 4: Analysis on the effectiveness of general-purpose MLLM jointly processing information. Condition means that how many models provide correct information for the general-purpose model, and such effectiveness is measured by the correct rate of decisions.

| Condition | Platform | | | Element Type | | | | | Overall |
|-----------|-----|---------|--------|--------|------|----------|-------|--------|---------|
| | Web | Desktop | Mobile | Button | Icon | Dropdown | Input | Toggle | |
| 1 model correct | 70.3 | 52.9 | 55.0 | 60.7 | 51.1 | 46.2 | 54.0 | 54.4 | 56.8 |
| 2 models correct | 82.6 | 88.9 | 87.4 | 84.3 | 84.1 | 86.2 | 88.2 | 93.6 | 87.0 |

interaction trajectory samples, featuring reasoning and reflection. UGround-V1-7B is uses Qwen2-VL-7B as backbone, mainly trained on grounding data, doing well on giving precise position of desired GUI elements. Utilizing relatively small scale MLLMs makes the agent system easier to deploy and improves its accessibility and availability.

## 4.2 EVALUATION RESULTS

The quantitative evaluation results of our method measured by success rate on UI-I2E-Bench and ScreenSpot benchmarks are shown in Tables. 1 and 2 respectively. We also compare our method with various recent GUI automation agents as baselines.

GAIR achieves the success rate of 78.4% on UI-I2E-Bench benchmark, significantly outperforming currently state-of-the-art baseline UGround-V1-72B with the success rate of 76.3% using no more than 30B of total parameters. In addition, GAIR maintains strong performance across various platform types and element types, achieving the highest performance in most circumstances. On ScreenSpot benchmark, GAIR reaches a success rate of 91.0%, surpassing competitive baselines such as UI-TARS-7B with 89.5% and UGround-V1-72B with 89.4%, with consistent high performance across various platforms. Such results demonstrates the effectiveness and reliability of GAIR framework.

## 4.3 ANALYSIS

We conducted analysis of GAIR's components on UI-I2E-Bench, where the results are shown in Tables. 3 and 4. Table. 3 is an ablation analysis, while Table. 4 shows the correct rate of decisions made by the general-purpose MLLM under different conditions, helping us to analyze the effectiveness of GAIR framework.

**Effectiveness of joint information reasoning.** Joint information reasoning enables the GAIR framework to combine strengths and integrate knowledge from different models. As demonstrated in Table. 3, joint information reasoning promotes the system's performance significantly. Compared to the 70.3% overall success rate of optimal single model, in this case UGround-V1-7B, utilizing multiple GUI-specific MLLMs raises the overall success rate to 76.5%. This indicates that the general-purpose MLLM is capable of processing information provided by the GUI-specific models, combining each model's strengths while complementing weaknesses on knowledge and capabilities. In addition, the correct decision rate of 56.8% and 87.0% when the information provided by 1 model and 2 models are correct respectively further indicates the information jointly processing abilities

of the general-purpose MLLM, with the numeric values significantly surpassing random selection. Therefore, the joint information reasoning is effective on combining information and capabilities from multiple models.

**Effectiveness of group reflection.** Group reflection mechanism offers GAIR system with an opportunity to avoid and correct wrong reasoning and decisions, which is crucial for reducing time latency and enhancing user experience in certain scenarios. The performance promotion shown in Table. 3 indicates that group reflection mechanism has helped GAIR to correct its inappropriate decisions in a lot of cases, improving the performance from 76.5% to 78.4%. The performance edge of GAIR to state-of-the-art models also indicates that group reflection mechanism can enhance the capabilities of agent systems rather than utilizing more parameters and data to construct stronger models. Thus, group reflection mechanism also shows a novel way of building more efficient and reliable agent systems when computational resources are limited. The group reflection mechanism introduced into GAIR framework successfully improves the performance of the framework by enabling error avoidance and correction.

**Effectiveness of GUI-specific MLLM.** GUI-specific models feature analyzing GUI screenshots and user requirements, extracting information from them, and give a precise operation based on the information. Such capabilities is the foundation and the basis of the whole framework. In addition, the GAIR framework employs multiple GUI-specific MLLMs, each with a unique vision encoder. Trained on different corpus and combined with the LLMs in different ways, these vision encoders provide views of GUI pages in multiple aspects. Such mechanism would further enhance GAIR's capabilities, for different aspects of views offered for different GUI-specific models could result in thoughts reflected by different model output, and output could get read by the general-purpose model capable of processing complex contextual information. Furthermore, different models are good at different types of tasks, enhancing GAIR's performance with a similar mechanism.

**Effectiveness of general-purpose MLLM.** As demonstrated in Tables. 3 and 4, general-purpose MLLM in GAIR framework features joint information processing and reasoning. Such capabilities promotes the framework's performance significantly, from 70.3% to 76.5%. The outstanding contextual and semantic information processing capabilities and the multi-turn dialogue mechanism enables GAIR system to efficiently integrate information from multiple models, thus combining knowledge and capabilities from multiple models. Furthermore, general-purpose model provides additional world knowledge for the agent framework, which would help with making correct decisions and correcting errors in particular circumstances. As demonstrated in Table. 1, GAIR framework's outstanding performance when processing tasks related with complex GUI elements such as *Input* and *Toggle* outperforming the second best by 4.1% and 5.8% respectively, where external world knowledge from the general-purpose model may make sense. Therefore, general-purpose MLLM enhances GAIR framework's capabilities by jointly processing information and introducing additional world knowledge.

## 5 CONCLUSION AND DISCUSSION

In this work, we present GAIR, a GUI automation agent framework that features combining capabilities and integrating knowledge from multiple heterogeneous models. By leveraging the strengths of multiple GUI-specific MLLMs and general-purpose MLLM, the agent system could gather sufficient information from GUI environment, process such information with deep reasoning and sound operation selection, and make further reflection when further information is necessary. By jointly processing information from multiple GUI-specific MLLMs and introducing group reflection mechanisms that drives the models to try to complete the task better and make use of the models' abilities more fully, thus reaching higher performance. Our evaluation on both benchmarks has validated such the effectiveness and reliability of GAIR framework.

However, GUI automation tasks in various vertical fields and particular scenarios are not yet well defined and sufficiently explored. If GUI-specific datasets and models in these professional fields get well constructed and deployed, we would be able to combine capabilities and integrate knowledge for multiple disciplines and scenarios by utilizing GAIR framework or other similar frameworks. Our future work will focus on exploring solutions of GUI automation tasks in specific scenarios and optimizing or upgrading GAIR framework accordingly. Such advancements will further improve GAIR's effectiveness and reliability while largely expanding the scope of GAIR's capabilities.

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

## A   LLM USAGE

LLM usage is limited in the scope of language polishing during our research work. We understand that we are ultimately responsible for the contents written under our name, including content generated by LLMs that could be construed as plagiarism or scientific misconduct.

