# OpenReview forum: "GAIR : GUI Automation via Information-Joint Reasoning and Group Reflection"
_ICLR.cc/2026/Conference — ICLR 2026 Conference Withdrawn Submission_

### Official Review · Reviewer_YHYh · 2025-10-22

**Soundness:** 3
**Presentation:** 1
**Contribution:** 2
**Rating:** 2
**Confidence:** 3

**Summary:**

This paper proposes a novel MLLM-based GUI automation agent framework named GAIR for integrating knowledge and combining capabilities from heterogeneous models to build GUI automation agent systems with higher performance. GAIR ensembles multiple GUI-specific MLLM to jointly process the GUI information and introduce a general-purpose MLLM to reason and making decisions. When the general-purpose model thinks that there is  insufficient information for a reasonable decision, GAIR would transit into group reflection, where the general purpose model would provide GUI-specific models with different instructions and hints based on their strengths and weaknesses, driving them to gather information with more significance and accuracy that can support deeper reasoning and decision. Experimental results on two grounding benchmarks demonstrate that GAIR can outperform single-model baselines.

**Strengths:**

1. The paper directly tackles the challenge of balancing specialization (GUI-specific models) and generalization (general-purpose models), which is a core issue in building versatile agents.
2. Integrating multiple GUI-specific MLLM to cape with different GUI tasks is an interesting idea, as different models have different strengths and weaknesses and can ensemble to deal with more complex and diverse tasks.
3. Experimental results demonstrate that GAIR can outperform single-model baselines on two grounding benchmarks.

**Weaknesses:**

1. Section 3 suffers from significant structural issues, with the majority of its content overly concentrated on background information at the expense of a clear and detailed exposition of the methodology. For example, in Section 3.1, Line 191-206 are overly redundant for the background, whereas the methodology part is not well-explained in the subsequent paragraph. While the actual ensemble mechanism of multiple GUI-specific MLLM is not well-explained. Similar issues exist in Section 3.2 and Section 3.3.
2. As the general-propose MLLM only accept textual perception results from GUI-specific MLLMs, is the multimodal capability of the general-purpose MLLM redundant? And what is the difference between this system and traditional textual-parsing-based agents for GUI automation?
3. Although the author claims that GAIR can achieve higher performance on GUI tasks, the experiments lie solely with grounding benchmarks. Therefore, the evaluation is insufficient.
4. The author set Qwen2.5-VL-7B as the general-propose MLLM in the agent system. However, this model is capable for GUI agentic tasks and could introduce bias into the comparisons.
5. On two grounding benchmarks, GAIR cannot consistently achieve the optimal results. Since the multi-MLLM collaboration framework could increase the computational cost and latency, it is necessary to investigate the trade-off between computational cost and performance.
6. Typos:
   - In Line 318, Section 4.1.2, the citation of Qwen2.5-VL-7B should be `citep` rather than `citet`

**Questions:**

1. Is the multimodal capability of the general-purpose MLLM actually used, or is it redundant under the current framework?
2. What is the difference between this system and traditional textual-parsing-based agents for GUI automation?
3. Add evaluations on agentic benchmarks such as AndroidControl and AndroidWorld.
4. Explain the minimal improvements on ScreenSpot between GAIR and UI-TARS-7B.

---

### Official Review · Reviewer_LgFb · 2025-10-28

**Soundness:** 3
**Presentation:** 3
**Contribution:** 3
**Rating:** 2
**Confidence:** 4

**Summary:**

This paper proposes GAIR, an MLLM‑based GUI agent framework that ensembles multiple GUI‑specialized models for perception/grounding and uses MLLM to jointly reason over their textual outputs, make an action decision, and enter a group reflection phase. In that phase, the general model emits model‑specific instructions/hints targeted at each GUI specialist’s strengths/weaknesses, gathers refined observations, and re‑reasons before executing an action.

Experiments on two offline grounding/interaction benchmarks, UI‑I2E‑Bench and ScreenSpot, show GAIR  achieves 78.4% success on UI‑I2E‑Bench and 91.0% on ScreenSpot, surpassing 72B‑scale baselines reported on public leaderboards while using ≤30B aggregate parameters. The ablation on p.7 shows (i) joint information reasoning improves over the best single model, and (ii) group reflection provides a further lift.

**Strengths:**

S1: The paper’s design, separating perception (GUI specialists) from integration and decision-making (a general VLM) is well-grounded in recent progress in GUI grounding (e.g., UGround, UI-TARS) and general VLM reasoning (e.g., Qwen2.5-VL). The motivation is clear, and the resulting architecture is both conceptually simple and practically meaningful.

S2: The approach achieves competitive results with relatively small models. On UI-I2E-Bench and ScreenSpot (Tables 1–2, p.6), GAIR surpasses 72B-parameter baselines while using only 3–7B specialists plus a 7B hub, demonstrating strong efficiency-performance tradeoffs that are valuable for practical deployment.

S3: The ablation studies (Tables 3–4) provide interpretable and quantitative insights into the benefits of joint information reasoning and group reflection. The analysis contrasting “1 model correct vs. 2 models correct” offers a clear and informative perspective on the aggregator’s robustness and reliability.

**Weaknesses:**

W1: The results are restricted to offline grounding scenarios. However, the community increasingly expects agentic evaluations in interactive OS, web, or mobile environments, with metrics such as task success, action steps, error recovery, and latency (e.g., Windows Agent Arena, AndroidWorld, WebArena, WebLINX, Mind2Web). Incorporating at least one such environment would greatly strengthen the paper’s claims about “GUI automation,” planning, and error correction capabilities.

W2: The paper does not clearly specify the triggering criterion for reflection, whether it relies on a confidence estimator, heuristic rule, or ensemble disagreement. Similarly, it remains unclear how specialist hints are derived and how failure modes (e.g., reflection loops or over-corrections) are mitigated. A detailed description of system inputs and outputs, reflection budgets (e.g., number of turns), and stopping conditions would improve reproducibility and understanding.

W3: The paper omits evaluation on ScreenSpot-Pro and ScreenSpot-v2. Including these datasets would significantly strengthen the empirical evidence supporting GAIR’s effectiveness.

W4: The proposed “group reflection” mechanism bears conceptual similarity to prior reflective paradigms such as Reflexion, Self-Refine, and ReAct. The paper should clarify what is novel about its approach, e.g., targeted per-specialist hinting or structured reflection orchestration and justify these design choices in contrast to existing frameworks.

**Questions:**

Q1: What specific criterion causes GAIR to enter reflection (e.g., entropy/uncertainty over action, inter‑model disagreement threshold, heuristic keywords like “not sure”)? Please provide an algorithm/pseudocode.

Q2: How are “strengths/weaknesses” of specialists estimated, fixed prior knowledge, validation‑time profiling by element type, or learned during inference? Please include examples of model‑specific prompts issued during reflection.

Q3:  Can you provide qualitative cases where reflection changed an initially wrong decision to right, and the reverse?

---

### Official Review · Reviewer_BUQM · 2025-10-31

**Soundness:** 3
**Presentation:** 3
**Contribution:** 2
**Rating:** 4
**Confidence:** 3

**Summary:**

This paper presents GAIR (GUI Automation via Information-Joint Reasoning and Group Reflection), a framework for GUI automation based on multiple multimodal large language models (MLLMs). GAIR integrates outputs from several GUI-specific models through a general-purpose MLLM that conducts information fusion and decision-making. When available information is insufficient, the system enters a group reflection phase to obtain additional data for further reasoning. Experimental results on UI-I2E-Bench and ScreenSpot indicate that GAIR achieves higher task success rates than previous models while using smaller MLLMs.

**Strengths:**

1.	The paper is well-written and free of notable grammatical errors.
2.	It achieves a significant performance improvement on existing datasets.

**Weaknesses:**

1.	The main issue of this paper is that the evaluation is conducted only on grounding benchmarks (UI-I2E-Bench and ScreenSpot), without testing the model in dynamic environments. Such dynamic evaluations can better reflect the model’s real GUI operation performance. The method should be validated on currently popular dynamic benchmarks such as AndroidWorld, OSWorld and Mobile-Agent.
2.	The proposed method lacks innovation, as both the multi-agent reasoning and reflection modules are commonly used in existing GUI automation approaches.
3.	In addition, multi-model reasoning increases inference time and token consumption, but the paper does not provide any quantitative analysis of these aspects.
4.	The paper does not include comparisons with the latest models, such as UI-Tars-2 and Mobile-Agent-v3.

**Questions:**

(1)	The paper would benefit from a more detailed description of the Group Reflection mechanism. Specifically, it would be helpful to clarify which models are employed, how many models are involved in simultaneous reasoning, and whether interactions occur among them.
(2)	It would be valuable to include visualization examples to help readers better understand the proposed mechanism and its effects.

---

### Official Review · Reviewer_QHCU · 2025-10-31

**Soundness:** 3
**Presentation:** 4
**Contribution:** 3
**Rating:** 4
**Confidence:** 3

**Summary:**

This paper proposes GAIR, a GUI automation agent framework that integrates multiple heterogeneous GUI-specific MLLMs with a general-purpose MLLM through two novel mechanisms: Information-Joint Reasoning and Group Reflection. The goal is to enable more generalizable and reliable GUI automation across diverse platforms (web, desktop, mobile) and application domains. GAIR leverages several specialized models (UI-TARS-7B-DPO, InfiGUI-R1-3B, UGround-V1-7B) for perception and a general MLLM (Qwen2.5-VL-7B) for reasoning, decision-making, and coordination. The Group Reflection mechanism allows the system to re-query and refine information when confidence is low, effectively enabling multi-turn self-correction. Experiments on UI-I2E-Bench and ScreenSpot demonstrate significant gains over prior GUI automation agents (e.g., +2.1% over UGround-V1-72B while using smaller models).

**Strengths:**

- Strong conceptual framing: The idea of combining multiple GUI-specialized MLLMs with a general reasoning model provides a pragmatic, scalable direction for GUI agents beyond single-backbone fine-tuning.
- Novel reflection design: The Group Reflection mechanism adds interpretability and robustness, addressing task incompleteness and model disagreement — a recurrent issue in multimodal agent pipelines.
- Comprehensive benchmarking: Evaluation across UI-I2E-Bench and ScreenSpot benchmarks includes ablation studies (single vs. multi-model vs. reflection) that clearly isolate the contribution of each component.
- Efficiency and performance: Achieving SOTA performance (78.4% / 91.0%) with only 7B-scale models highlights an attractive compute-performance tradeoff.

**Weaknesses:**

- Limited technical depth in integration mechanics: The “joint reasoning” interface between GUI-specific and general MLLMs is conceptually sound but not clearly formalized (e.g., message passing format, fusion schema, or reasoning trace examples are absent).
- Reflection trigger ambiguity: It remains unclear how the "insufficient information" criterion is operationalized—whether through confidence scores, natural-language self-assessment, or heuristic thresholds.
- Scalability and latency: Running several MLLMs in sequence may cause high inference cost; parallelization and runtime efficiency are not discussed.
- Incomplete ablations. The study omits reflection-only or joint-reasoning-only baselines, making it hard to isolate the contribution of each component.

**Questions:**

- How is information fusion implemented between the GUI-specific MLLMs and the general-purpose MLLM? Are the outputs concatenated textually, ranked, or structured via a schema before being fed into Qwen2.5-VL?
- The framework currently integrates three GUI-specialized models. Would adding more models linearly improve performance or saturate quickly due to redundancy?
- Have the authors analyzed failure cases where reflection did not improve or worsened results? Such insights could reveal limits of self-correction and help guide adaptive stopping strategies.

---

### Note · Authors · 2025-12-02

I have read and agree with the venue's withdrawal policy on behalf of myself and my co-authors.